# Immune-Checkpoint Inhibitors for Metastatic Colorectal Cancer: A Systematic Review of Clinical Outcomes

**DOI:** 10.3390/cancers13174345

**Published:** 2021-08-27

**Authors:** Dmitrii Shek, Liia Akhuba, Matteo S. Carlino, Adnan Nagrial, Tania Moujaber, Scott A. Read, Bo Gao, Golo Ahlenstiel

**Affiliations:** 1Blacktown Clinical School, Western Sydney University, Sydney, NSW 2148, Australia; Dmitri.Shek@health.nsw.gov.au (D.S.); S.Read@westernsydney.edu.au (S.A.R.); 2Blacktown Hospital, Sydney, NSW 2148, Australia; Matteo.Carlino@health.nsw.gov.au (M.S.C.); Adnan.Nagrial@health.nsw.gov.au (A.N.); Tania.Moujaber@health.nsw.gov.au (T.M.); Bo.Gao@health.nsw.gov.au (B.G.); 3Accreditation Centre, RUDN University, 117198 Moscow, Russia; Akhuba_lg@rudn.university; 4Crown Princess Mary Cancer Centre, Westmead Hospital, Sydney, NSW 2145, Australia; 5Westmead Clinical School, The University of Sydney, Sydney, NSW 2145, Australia; 6Melanoma Institute Australia, Sydney, NSW 2065, Australia; 7Storr Liver Centre, Westmead Institute for Medical Research, Sydney, NSW 2145, Australia

**Keywords:** colorectal cancer, pembrolizumab, atezolizumab, nivolumab, ipilimumab, targeted therapy, immune-checkpoint inhibitors

## Abstract

**Simple Summary:**

In 2018, colorectal cancer (CRC) was declared the fourth most deadly cancer worldwide. Unfortunately, a quarter of all patients are diagnosed at late stages, when curative surgery is not possible, rendering systemic therapy and/or best supportive care as the only options. To our knowledge, this is the first systematic review assessing response and survival rates in patients with mCRC treated with immune-checkpoint inhibitors (ICIs). Our study established that ICIs show potentially superior response rates in mCRC patients, though only in those with high microsatellite instability (MSI). Nivolumab + regorafenib was reported to provide encouraging response in low-MSI (MSI-L) patients; however, additional studies using cohort randomization are required. Further studies are required, particularly regarding the mechanism of resistance to ICIs in MSI-L patients.

**Abstract:**

Background. Colorectal cancer (CRC) is the fourth most deadly cancer worldwide. Unfortunately, a quarter of the patients are diagnosed at late stages, when surgical options are limited. Targeted therapies, particularly immune-checkpoint inhibitors (ICIs), are the latest addition and have been studied herein regarding their efficacy outcomes. Methods. Clinical studies were identified through the PubMed, Scopus and Cochrane databases. Any trial that evaluated ICIs in patients with metastatic CRC (mCRC) and reported the objective response rate was deemed eligible. Data analysis was performed by employing the random-effects model in STATA v.17. Results. A total of 461 articles were identified; 13 clinical trials were included, encompassing a total cohort of 1209 patients. Our study determined that a single PD-1/PD-L1 checkpoint blockade provides durable clinical response in mCRC patients with high microsatellite instability (MSI-H). The combinatorial therapy of CTLA-4 + PD-1 inhibitors also showed high response rates in pre-treated MSI-H patients. The single-arm REGONIVO trial reported durable clinical response in patients with microsatellite stable (MSS) status. Conclusions. Our study surmises that PD-1/PD-L1 inhibitors as well as combination therapy with CTLA-4 and PD-1 inhibitors show encouraging response rates in mCRC patients, albeit exclusively in patients with cancer that are of MSI-H status. A single study suggests that nivolumab + regorafenib can reach a durable response rate in MSS patients; however, further studies in larger randomized settings are required.

## 1. Introduction

Colorectal cancer (CRC) is the fourth most common cause of cancer-related mortality, accounting for 10% of all annually diagnosed malignancies worldwide [1]. It is the second most common cancer in women and the third most common in men [2], with the yearly incidence to surpass 2.5 million by 2035 [3]. Despite the availability of screening programs, 25% of all new CRC cases are still diagnosed at advanced stages [4,5] (Figure 1). For decades, chemotherapy has been the primary treatment for metastatic CRC [6], particularly fluorouracil (5-FU/F), oxaliplatin (OX), irinotecan (IRI) and capecitabine (CAP), alone or in combination [7]. Unfortunately, high risk of systemic toxicity, poor response rates and low efficacy have driven the search for better treatment options with higher tumor-specific selectivity, especially among patients with late-stage disease progression [7].

Targeted therapies provide an alternative for patients with metastatic CRC. Targeted therapies function by blocking specific molecules involved in cancer growth and dissemination [9]. In CRC, several potential targets have been studied over the past 20 years (Figure 2), including epidermal growth factor receptor (EGFR) inhibition [10] as well as suppression of the Ras–Raf–MEK–ERK pathway, which is responsible for cell growth, proliferation and survival [11,12]. Vascular endothelial growth factor-A (VEGF-A), which promotes tumor angiogenesis [13,14,15], is another critical target. The first anti-VEGF-A mAb, bevacizumab, was approved in 2004 [16]. Since then, the Food and Drug Administration (FDA) has granted approval to aflibercept (inhibitor of VEGF-A), ramucirumab (fully humanized mAb against VEGFR-2) and regorafenib (inhibitor of VEGFR-2) for the treatment of CRC [16]. However, treatment resistance is inevitable and novel treatment options are needed. 

A recent therapeutic addition to the CRC treatment regimen are immune-checkpoint inhibitors (ICIs) [17]. ICIs are mAbs targeting inhibitors of T cell receptor (TCR) activation, particularly programmed cell death-1 (PD-1), programmed cell death ligand 1 (PD-L1) and cytotoxic T lymphocyte-associated antigen-4 (CTLA-4) expressed in T cells and antigen-presenting cells [17]. They have become widely used in many cancers, including melanoma, RCC, NSCLC and HCC [18]. In mCRC, this treatment modality has already shown an encouraging clinical response [19] in patients with high microsatellite instability (MSI-H) in mismatch repair (MMR) genes such as MutL homolog 1 (*MLH1*), MutS homolog 2, 6 (*MSH2*, *MSH6*), *PMS2* and tumor-associated calcium signal transducer 1/epithelial cell adhesion molecule (*TACSTD1/EPCAM*) [20]. However, MSI-H CRC accounts for 5% of the total cases, with the remaining 95% of CRC cancers classified as microsatellite stable (MSS) or low microsatellite instability (MSI-L) [20], where the efficacy of ICIs is yet to be defined. This study aims to comprehensively summarize existing knowledge regarding response and survival rates and outline major barriers to ICI treatment in patients with mCRC.

## 2. Methods

### 2.1. Eligibility Criteria and Data Collection

For the purpose of this study, we utilized the following eligibility criteria: (1) patients with mCRC; (2) PD-1, PD-L1, CTLA-4 inhibitors used for treatment either as a single agent or in combination with chemotherapeutic/biological agents; (3) ORR was reported; (4) published in English. Systematic reviews and meta-analysis, case reports, studies with no ORR data and studies not published in the English language were excluded.

The following data were obtained from studies deemed eligible: (1) NCT, trial phase; (2) number of patients; (3) treatment regimen(s); (4) number of previous lines of treatments; (5) ORR with 95% CI; (6) median PFS with 95% CI; (7) microsatellite instability (MSI) status (Table 1).

### 2.2. Study Endpoints

The primary endpoint is to systematically determine the ORR (according to Response Evaluation Criteria in Solid Tumors version 1.1 [RECIST v1.1]) in patients with mCRC treated with ICI-based therapies. In addition, we summarized median progression-free survival (PFS) with 95% confidence interval (CI), if reported across selected studies.

### 2.3. Literature Search and Study Endpoints

This systematic review is reported in accordance with PRISMA (Preferred Reporting Items for Systematic Reviews and Meta-Analyses) reporting guidelines and is registered at the PROSPERO database (CRD42020197617) [33].

Three authors (D.S., L.A. and G.A.) independently conducted a comprehensive literature review through the electronic databases Scopus, MEDLINE, Web of Science and Cochrane Library from 30 September 2004 to 30 April 2021 using keywords (*immune-checkpoint inhibitors, metastatic colorectal cancer, colon cancer, rectal cancer, targeted therapy, monoclonal antibodies, pembrolizumab, nivolumab, ipilimumab, atezolizumab*) linked by operators “AND” and “OR.” Abstracts presented at annual meetings of the American Society of Clinical Oncology (ASCO) and the European Society for Medical Oncology (ESMO) were also examined. Abstracts were considered eligible if the study was published for the first time and matched eligibility criteria. Finally, we searched through bibliographies of selected articles as well as clinical trial registries (www.clinicaltrials.gov (accessed from 1 May 2021 to 25 May 2021) using the aforementioned keywords.

The records found through primary search were initially screened by title and abstract. The full text of potentially eligible studies was reviewed, and if eligible the study was included in the analysis. Selected studies were reviewed by all authors and all discrepancies were solved by consensus. If one study was reported multiple times, the study with the most comprehensive and up to date data was included. 

### 2.4. Data Analysis

Risk of bias in randomized studies was assessed using a revised Cochrane tool to assess risk of bias in randomized trials [34,35] whereas single-arm studies were assessed by the Newcastle-Ottawa tool for Cohort Studies [36]. PFS is defined as time (in months) from randomization to recurrence [37]. Study heterogeneity was calculated with *I*^2^. *I*^2^ values less than 25%, 50% and 75% were considered to be of low, moderate and high study heterogeneity, respectively. The random-effects model (DerSimonian and Laird procedure) was applied to investigate the best therapeutic outcome among the selected studies. Results were deemed statistically significant if *p*-level <0.05. Data synthesis was performed using Stata version 17 (Stata Corp. LLC) software. 

## 3. Results

A total of 461 articles were identified using the defined keywords, of which 13 clinical trials were included in the qualitative and quantitative analyses based on the outlined eligibility criteria with a total cohort of 1209 patients. The PRISMA flow chart is shown in Figure 3.

### 3.1. Checkpoint Blockade in Colorectal Cancer Patients with High Microsatellite Instability

#### 3.1.1. Single PD-1 Inhibitor

KEYNOTE-177 (NCT02563002), a randomized phase 3 clinical trial, investigated the efficacy of first-line pembrolizumab (PEMBRO 200 mg Q3W) in patients with MSI-H/deficient mismatch repair (dMMR) CRC compared to the standard of care (SoC) FOLFIRI (folinic acid, fluorouracil, irinotecan) or FOLFOX6 (folinic acid, fluorouracil, oxaliplatin), +/− bevacizumab or cetuximab [19]. The median PFS reached 16.5 months in the PEMBRO-treated group (*n* = 153) compared to 8.2 months in the chemotherapy (*n* = 154) group (HR: 0.5 (0.45–0.8), *p* = 0.0002) [19]. PEMBRO-treated ORR reached 43.8% (95% CI: 35.8 to 52), with 11.1% (*n* = 17) and 32.7% (*n* = 50) of the examined patients with complete (CR) and partial responses (PR), respectively. In total, 67% (*n* = 102) and 30% (*n* = 42) had right- and left-side colon malignancy, respectively, and 75% (*n* = 115) did not receive any prior therapy. The study concluded that PFS is consistently longer in patients treated with PEMBRO and resulted in the recent update of therapeutic guidelines for mCRC patients with MSI-H/dMMR [38].

A phase 2 single-arm trial, KEYNOTE-164 (NCT02460198), determined the efficacy of PEMBRO 200 mg Q3W in mCRC patients with MSI-H status previously treated with ≥2 (cohort A, *n* = 61) and ≥1 (cohort B, *n* = 63) lines of systemic therapies [22]. The median PFS reached 2.3 months (95% CI: 2.1 to 8.1) and 4.1 months (95% CI: 2.1 to 18.9) in cohorts A and B, respectively [22]. The ORR in both cohorts was 33%, including 2 CRs and 18 PRs in cohort A and 5 CRs and 16 PRs in cohort B [22]. The data regarding tumor location were not reported. Finally, the authors concluded that PEMBRO provides durable clinical response in MSI-H/dMMR patients progressed on ≥2 and ≥1 lines of previous systemic treatment [22].

With regard to nivolumab (NIVO) monotherapy, the CheckMate142 (NCT02060188) phase 2 single-arm study established an ORR of 32% (95% CI: 22 to 44) with the median PFS of 14.3 months (95% CI: 4.3 to NR) [23]. Whereas 99% (*n* = 73) of the patients had at least one line of prior chemotherapy, 100% of the patients were MSI-H/dMMR [23]. The authors concluded that NIVO 3 mg/kg Q2W reached a durable clinical response and survival rates and could be considered in further clinical trials; however, the primary tumor location was not reported [23]. 

#### 3.1.2. Single PD-L1 Inhibitor

A phase 1b trial (NCT01633970) investigated atezolizumab (ATEZO) + various chemotherapeutic/biological (bevacizumab) regimens in patients with advanced solid tumors [24] and recorded an ORR in mCRC patients of 30% (95% CI: 6.7 to 65.3), with the median PFS not reached by the time of data cutoff [24]. ATEZO 10 mg/kg Q3W was used in combination with bevacizumab 15 mg/kg Q3W among *n* = 10 enrolled patients, of whom *n* = 7 and *n* = 3 had ≥2 and 1 line(s) of previous chemotherapy, respectively [24]. Therapeutic efficacy was encouraging, and follow-up is ongoing, with the final results not yet reported. 

A phase 2 single-arm study (NCT03150706) established that avelumab 10 mg/kg Q2W resulted in an ORR of 24.2% (12.1% (*n* = 4) and 12.1% (*n* = 4) achieved CR and PR, respectively) in MSI-H mCRC patients [29]. The median PFS reached 3.9 months (95% CI: 2.3 to 5.6) [29]. The primary tumor was located in the right and left sides in 66.7% (*n* = 22) and 15.2% (*n* = 5) of patients, respectively [29]. A total of 48.5% (*n* = 16) and 51.5% (*n* = 17) received >1 and >2 lines of previous chemotherapy, respectively. Overall, avelumab displayed a durable clinical response in MSI-H mCRC patients progressed on standard chemotherapy.

A phase 2 single-arm study (NCT01693562) investigating durvalumab (DURVA) in patients with advanced solid tumors demonstrated an ORR of 22% (95% CI: 10 to 39) and a median PFS of 6 months (95% CI: 3 to 20) in *n* = 36 mCRC patients [25]. DURVA 10 mg/kg Q2W showed promising antitumor activity in pre-treated mCRC patients, with the final results of this trial anticipated by Fall 2021. Finally, the examined studies showed that PD-L1 inhibitors can also provide a durable clinical response in mCRC patients with MSI-H status. 

#### 3.1.3. Combination of CTLA-4 + PD-1 Inhibitors

CheckMate142 (NCT02060188), a phase 2 single-arm study, demonstrated encouraging efficacy of NIVO 3 mg/kg in combination with ipilimumab (IPI) 1 mg/kg in MSI-H mCRC patients. ORR reached 54.6% (95% CI: 45.2 to 63.8) with median PFS not reached by the time of data cutoff [21]. Primary tumor location was in the right and left side in 55% (*n* = 65) and 25% (*n* = 30) of the patients, respectively [21]. The 12-month PFS rate was 71% (95% CI:61.4 to 78.7) [21].

To summarize, the aforementioned trials validated ICIs as a therapeutic alternative for mCRC patients with MSI-H, particularly PEMBRO. NIVO as monotherapy or in combination with IPI has shown encouraging response rates and has also been approved by the FDA for mCRC patients progressed on standard chemotherapy. Finally, PD-L1 inhibitors have also shown encouraging response and survival rates; however, larger randomized trials are required (Figure 4).

### 3.2. Checkpoint Blockade in Colorectal Cancer Patients with Stable Microsatellite Status

#### 3.2.1. Immune-Checkpoint Inhibitors + Chemotherapeutic or Biological Agents

A phase 1b single-arm REGONIVO study (NCT03406871) investigated NIVO 3 mg/kg Q2W in combination with the multikinase inhibitor regorafenib 80–160 mg in patients with advanced gastric cancer or CRC [27]. It demonstrated an ORR of 36% (95% CI: 18 to 57.5) with a median PFS of 7.9 months (95% CI: 2.0 to NR) in the mCRC cohort (*n* = 25) [27]. Of note, 96% (*n* = 24) of the participants had MSS status with 20% (*n* = 5) and 80% (*n* = 20) had 2 and ≥3 lines of prior chemotherapy, respectively [27]. In 80% (*n* = 20) of the patients, the primary tumor was located in the left side or the rectum [27]. Overall, Fukuoka et al. reported encouraging antitumor activity in MSS mCRC patients, which are currently deemed ineligible for checkpoint blockade therapies. Further studies should be conducted to determine the outcomes of ICIs and kinase inhibitors in MSS patients.

A phase 3 randomized controlled trial, IMblaze 370 (NCT02788279), examined ATEZO 840 mg Q2W + cobimetinib 60 mg (cohort A) or ATEZO 1200 mg Q3W (cohort B) compared to regorafenib 160 mg in mCRC patients with MSS status [30]. The primary endpoint was not observed with no significant difference in overall survival compared to the control group treated with regorafenib [30]. A total of 93% (*n* = 170) and 92% (*n* = 83) of the patients in cohorts A and B, respectively, had MSS status [30]. Of note, the earlier trial (NCT01988896, phase 1b) also established a low clinical efficacy in MSS patients treated with ATEZO + cobimetinib [28]. These data have underscored the challenge of exploring ATEZO-based regimens in mCRC patients with MSS status.

A phase 2 randomized, placebo-controlled trial BACCI (NCT02873195) compared the outcomes between capecitabine + bevacizumab + placebo (arm A, *n* = 46) and capecitabine + bevacizumab + ATEZO (arm B, *n* = 82) in mCRC patients [26]. A total of 85.7% (*n* = 70) of patients had MSS status [26]. No significant differences between study and control groups were established [26]. Nonetheless, study investigators have decided to conduct a phase 3 trial to further examine this regimen in mCRC patients with MSS status.

#### 3.2.2. Combination of CTLA-4 + PD-L1 Inhibitors

A phase 2 randomized trial (NCT02870920) investigating DURVA + tremelimumab in mCRC patients with MSS status compared to the best supportive care (BSC) demonstrated no CR in the study group. In addition, only one patient (total cohort included *n* = 119) achieved PR with the median PFS of 1.8 months [32]. No significant differences in PFS between the studied regimens were observed; however, stable disease (SD) was recorded in 22.7% (*n* = 27) and 6.6% (*n* = 4) of the patients treated with DURVA-based regimen and BSC, respectively [32]. As the patients were of MSS status and had received at least one line of prior chemotherapy, the authors concluded that DURVA + tremelimumab may be of potential benefit for this patient population due to the lack of other treatment options [32].

To summarize, ICIs did not show encouraging clinical efficacy in mCRC patients with MSS status. The mechanisms underlying this resistance are currently unknown.

### 3.3. Study Limitations

Our study has limitations worth outlining. First, some of the selected studies did not define the number of therapeutic lines used prior to ICIs. Secondly, our analysis focused on determining efficacy clinical outcomes, disregarding safety outcomes in mCRC patients. 

## 4. Discussion

Targeted therapy has opened a new chapter in the management of CRC, with ICIs providing a new hope for patients with advanced colonic malignancies [39]. In particular, the humanized anti-PD-1 mAb PEMBRO has recently become a new first-line therapeutic alternative for mCRC patients with MSI-H status [19]. The randomized KEYNOTE-177 trial reported significantly higher PFS and ORR among patients treated with PEMBRO compared to patients treated with SoC chemotherapy (mFOLFOX6 or FOLFIRI +/− bevacizumab or cetuximab) [19]. PEMBRO has also shown high efficacy in pre-treated (>1 and >2 lines of standard chemotherapy) mCRC patients with MSI-H status [22]. Another single PD-1 inhibitor, NIVO, has also shown durable clinical response in pre-treated mCRC patients with MSI-H status [23]. Similarly, PD-L1 inhibitors (ATEZO, Avelumab and DURVA) also demonstrated encouraging response rates in pre-treated MSI-H mCRC patients in early phases of clinical trials, with further studies ongoing [24,25,29]. These results emphasized the higher efficacy of single ICIs in mCRC patients with MSI-H status. In addition to these findings, the CheckMate142 trial reported that NIVO + IPI reached an impressive response rate in heavily pre-treated mCRC patients with MSI-H status (99% had ≥1 lines of chemotherapy) [21]. Further investigation of doublet ICI therapy is currently ongoing.

By contrast, PD-1/PD-L1 inhibitors showed no clinical benefit in patients with MSS status [26,28,30,32]. The factors impacting lower response rates of ICI therapy in CRC patients with MSS status remain unknown. An analysis conducted by Mlecnik et al. [40] revealed that patients with MSI-H status possess a higher rate of mutations in *ACVR2A* (activin A receptor type 2A)*, FBXW7* (F-box and WD repeat domain-containing 7) and *CTNNB1* (catenin beta 1) genes, and fewer mutations in *APC* (adenomatous polyposis coli), *KRAS* and *TP53* (tumor protein 53) genes compared to patients with MSS status [41]. The underlying mechanisms to explain these differences are unknown. Other studies reported that MSI-H patients possessed higher tumor infiltration of CD3+, CD8+ and CD45RO+ T cells compared to their MSI-L counterparts [42,43]. Unfortunately, clinical trials using ICIs in patients with mCRC do not commonly conduct genome sequencing or phenotyping of tumor-infiltrating lymphocytes (TILs); thus, it is difficult to establish the impact of tumor mutational burden (TMB) or TILs across MSS patients within the selected clinical trials. The encouraging outcomes of REGONIVO trial (NIVO + regorafenib) [27] could perhaps be due to known activity of regorafenib to modulate the tumor immune microenvironment via polarization of antigen-presenting cells, particularly macrophages [44]. Ou et al. established that regorafenib is capable of inducing the p38MAPK/Creb1/Klf4 signaling pathway responsible for the activation of tumor-associated macrophages [44], and subsequent production of pro-inflammatory cytokines (IL-10, IL-12 and IL-23) responsible for activation of cytotoxic T cells [45]. Following on these insights, it is of particular interest to examine combinatorial regimens of ICIs with kinase inhibitors. Lastly, the expression of other checkpoint molecules (ICOS, TIM-3, LAG-3, GITR and OX40) and their role in T cell inhibition or activation in the context of CRC may explain the resistance to currently used PD-1/PD-L1 inhibitors [20]. Although the efficacy of novel checkpoint inhibitors is yet to be fully defined, there is hope that in the near future, the majority of CRC patients may be benefited from ICI-based therapies.

### Conclusion and Future Perspectives

Our analysis suggests that PD-1 and PD-L1 inhibitors may have significant potency in mCRC patients with MSI-H/dMMR status. FDA approval has already been granted to some regimens for use as therapeutic alternatives in aforementioned patient populations. Nonetheless, further evaluation in larger phase 3 clinical trials is necessary and is ongoing (Table 2). Despite the promising efficacy of checkpoint blockade, it is commonly regarded as ineffective in patients with MSS status. Nonetheless, NIVO + regorafenib shows promise, though the trial was non-randomized and in a small cohort. Although clinical trials determining novel ICI-based combinatorial regimens in MSS patients are ongoing, it is also critical to conduct specifically designed translational studies to establish precise mechanisms of resistance in these patients. These studies may reveal previously unknown molecular patterns of CRC in patients with MSI-L and provide new strategies for overcoming therapeutic barriers. Understanding these roadblocks may, one day, result in successful and effective implementation of ICI therapy in a majority of patients with severely progressed or refractory colorectal cancer.

## Figures and Tables

**Figure 1 cancers-13-04345-f001:**
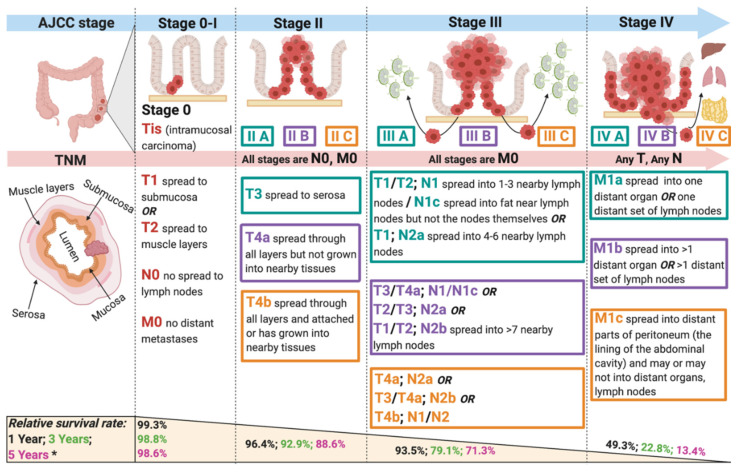
Colorectal cancer stages according to the American Joint Committee on Cancer (AJCC), 8th edition [8]. TNM staging is based on the size of the tumor, growth into lymph nodes and distant metastases to organs and/or tissues. Relative survival rate is an epidemiological characteristic comparing people with the specific histological type and stage of cancer to the overall population in a specific area/region of our country. 1-, 3- and 5-year relative survival rates reduce drastically as CRC progresses from stage 0 to 4.

**Figure 2 cancers-13-04345-f002:**
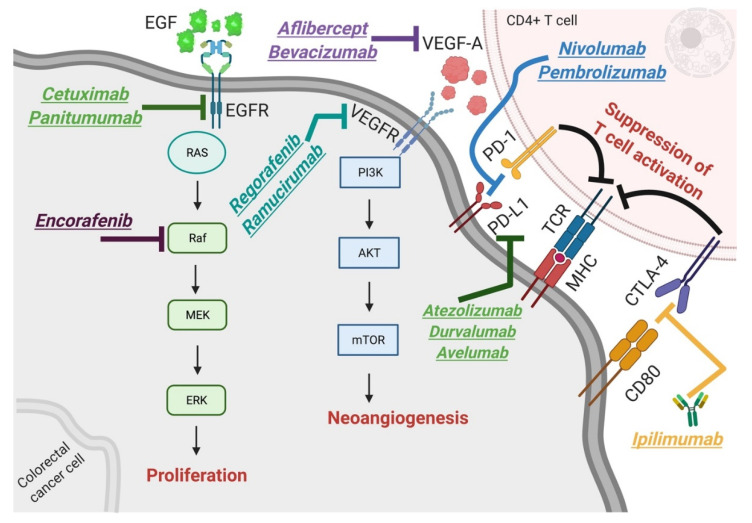
Molecular mechanisms of targeted drugs approved by the Food and Drug Administration for the treatment of metastatic colorectal cancer. (1) Anti-EGFR (epidermal growth factor receptor) monoclonal antibodies (mAbs) cetuximab and panitumumab; (2) encorafenib, an inhibitor of the Raf protein as part of the MAPK/ERK signaling pathway; (3) VEGF-A (vascular endothelial growth factor-A) inhibitors aflibercept and bevacizumab; (4) VEGF receptor inhibitors regorafenib and ramucirumab; (5) anti-PD-1 (programmed cell death-1) mAbs nivolumab and pembrolizumab and anti-PD-L1 (programmed cell death ligand 1) mAbs atezolizumab, durvalumab and avelumab; (6) anti-CTLA-4 (cytotoxic T lymphocyte-associated antigen-4) mAb ipilimumab. Current targeted therapies inhibit three major processes crucial for cancer growth: unrestricted proliferation, neo-angiogenesis and suppression of T cell immune responses.

**Figure 3 cancers-13-04345-f003:**
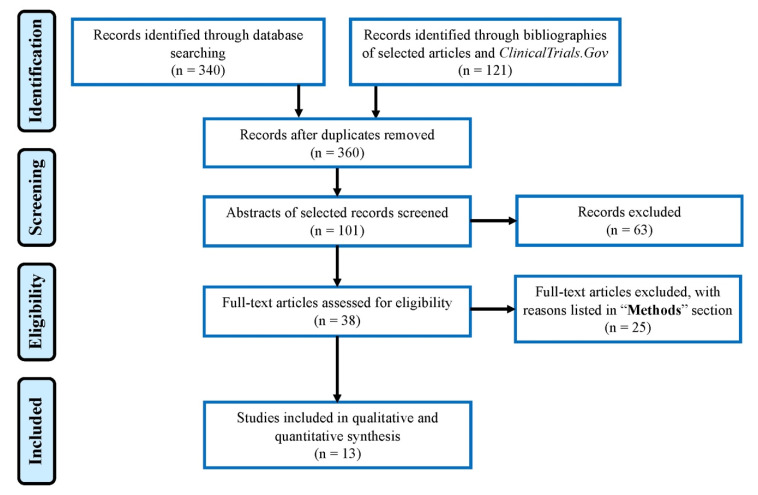
PRISMA flow chart of the systematic review.

**Figure 4 cancers-13-04345-f004:**
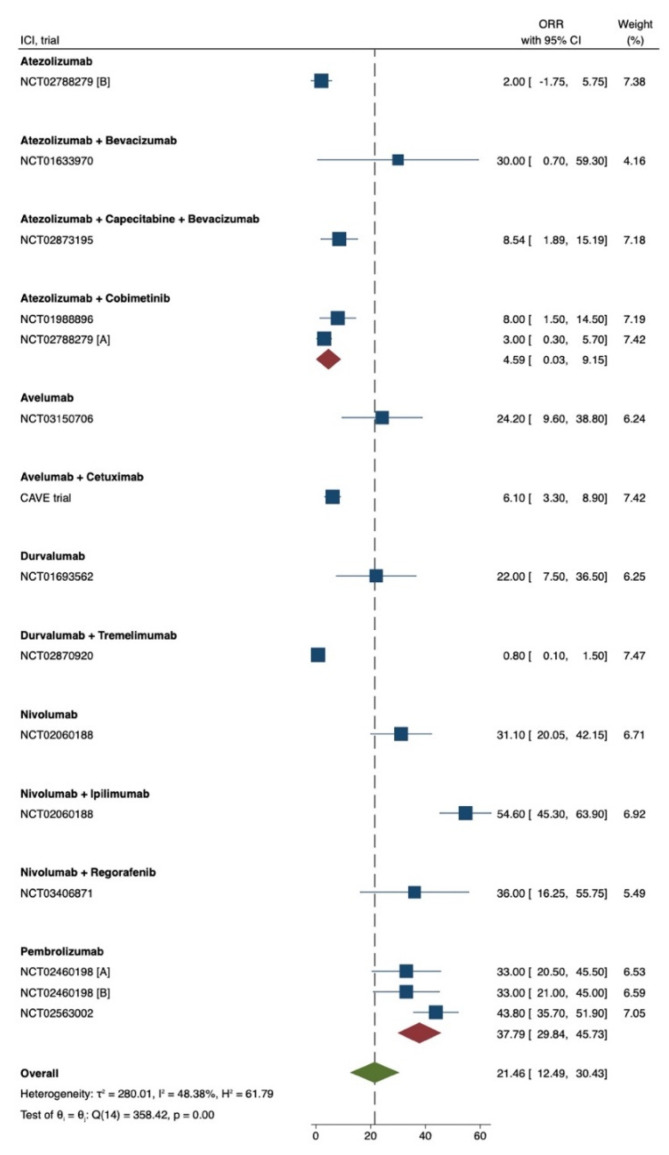
Forest plot visualizing the objective response rate across selected studies. The cumulative response rate is estimated to be 21.46% with non-significant differences in weight distribution across selected studies and study heterogeneity of 48.38% (deemed as moderate).

**Table 1 cancers-13-04345-t001:** List of studies included in this systematic review. ICI—immune-checkpoint inhibitor; NIVO—nivolumab; IPI—ipilimumab; PEMBRO—pembrolizumab; ATEZO—atezolizumab; DURVA—durvalumab; TREM—tremelimumab; Q3W—every 3 weeks; CI—confidence interval; PFS—progression-free survival; ORR—objective response rate; PD-L1—programmed cell death-1 ligand 1; mo—months; MSI-H—microsatellite instability-high; MSS—microsatellite stable; d/pMMR—deficient/proficient mismatch repair gene; wt—wild type; mt—mutated type.

NCT	Phase	ICI	Total N	Prior Systemic Treatment: % (*n*)	Median PFS (95% CI)	ORR (95% CI)	Marker Status: % (*n*)
NCT02060188 [21]	2	NIVO 3 mg/kg + IPI 1 mg/kg Q3W (4 doses) followed by NIVO 3 mg/kg Q2W	119	0: 1 (1) 1: 23 (27) 2: 36 (43) ≥3: 40 (48)	NR	54.6% (45.2 to 63.8)	*BRAF/KRAS* wt: 26 (31) *BRAF* mt: 24 (29) *KRAS* mt: 37 (44) PD-L1 ≥ 1%: 22 (26) PD-L1 < 1%: 55 (65) Unknown: 24 (28)
NCT02460198—cohort A [22]	2	PEMBRO 200 mg Q3W	61	1: 10 (6) 2: 46 (28) ≥3: 44 (27)	2.3 mo. (2.1 to 8.1)	33% (21 to 46)	MSI-H/dMMR: 100 (61) *BRAF/KRAS/NRAS* wt: 18 (11) *BRAF* mt: 15 (9) *KRAS* mt: 26 (16) *NRAS* mt: 5 (3)
NCT02460198—cohort B [22]	2	PEMBRO 200 mg Q3W	63	1: 38 (24) 2: 32 (20) ≥3: 30 (19)	4.1 mo. (2.1 to 18.9)	33% (22 to 46)	MSI-H/dMMR: 100 (63) *BRAF/KRAS/NRAS* wt: 9 (6) *BRAF* mt: 8 (5) *KRAS* mt: 35 (22) *NRAS* mt: 8 (5)
NCT02060188 [23]	2	NIVO 3 mg/kg Q3W	74	0: 1 (1) 1: 15 (11) 2: 30 (22) ≥3: 54 (40)	14.3 mo. (4.3 to NR)	31.1% (20.8 to 42.9)	MSI-H/dMMR: 100 (74) *BRAF/KRAS* wt: 39 (29) *BRAF* mt: 16 (12) *KRAS* mt: 35 (26) Unknown: 9 (7) PD-L1 ≥ 1%: 28 (21) PD-L1 < 1%: 64 (47) Unknown: 8 (6)
NCT01633970 [24]	1b	ATEZO 1200 mg Q3W + Bevacizumab	10	1: 30 (3) ≥2: 70 (7)	NR	30% (6.7 to 65.3)	MSI-H/dMMR: 100 (10)
NCT01693562 [25]	1/2	DURVA 10 mg/kg Q2W	36	-	6 mo. (3 to 20)	22% (10 to 39)	MSI-H/dMMR: 100 (36)
NCT02873195 [26]	2	Capecitabine + ATEZO 1200 mg Q3W + Bevacizumab	82	-	4.4 mo. (4.1 to 6.4)	8.54% (3.5 to 16.8)	MSS/pMMR: 85.7 (70)
NCT03406871 [27]	1b	NIVO 3 mg/kg Q2W + Regorafenib	25	-	7.9 mo. (2.9 to NR)	36% (18 to 57.5)	MSI-H/dMMR: 0 (0) MSS/pMMR: 100 (25) PD-L1 ≥ 1%: 42 (10) PD-L1 < 1%: 58 (14)
NCT01988896 [28]	1/1b	ATEZO 800 mg Q2W + cobimetinib	84	-	1.9 mo. (1.8 to 2.3)	8% (3 to 16)	MSI-H/dMMR: 2.4 (2) MSS/pMMR: 72 (60) MSI-Unknown: 21 (18) *BRAF* mt: 6 (5) *KRAS* mt: 68 (57) Unknown: 23 (19)
NCT03150706 [29]	2	Avelumab 10 mg/kg Q2W	33	1: 48.5 (16) 2: 33.3 (11) ≥3: 18.2 (6)	3.9 mo. (2.3 to 5.6)	24.2% (9.4 to 38.6)	*BRAF* mt: 12.1 (4) *KRAS* mt: 60.6 (20) Unknown: 27 (9)
NCT02563002 [19]	3	PEMBRO 200 mg Q3W	153	0: 75 (115) >1: 25 (38)	16.5 mo. (5.4 to 32.4)	43.8% (35.8 to 52)	MSI-H/dMMR: 100 (153) *BRAF/KRAS* wt: 22 (34) *BRAF* mt: 22 (34) *KRAS* mt: 22 (34) Unknown: 34 (52)
NCT02788279—cohort A [30]	3	ATEZO 840 mg Q2W + cobimetinib	183	<3: 73 (134) >3: 27 (49)	1.91 mo. (1.87 to 1.97)	3% (0.9 to 6.3)	MSI-H/dMMR: 2 (3) MSS/pMMR: 93 (170) *BRAF* wt: 95 (174) *KRAS* wt: 46 (84) *BRAF* mt: 5 (9) *KRAS* mt: 54 (99) PD-L1 ≥ 1%: 43 (79) PD-L1 < 1%: 46 (84) Unknown: 11 (20)
NCT02788279—cohort B [30]	3	ATEZO 1200 mg Q3W	90	<3: 71 (64) >3: 29 (26)	1.94 mo. (1.91 to 2.1)	2% (0.3 to 7.8)	MSI-H/dMMR: 3 (3) MSS/pMMR: 92 (83) *BRAF* wt: 97 (87) *KRAS* wt: 46 (41) *BRAF* mt: 3 (3) *KRAS* mt: 54 (49) PD-L1 ≥ 1%: 39 (35) PD-L1 < 1%: 47 (42) Unknown: 14 (13)
CAVE trial [31]	2	Avelumab 10 mg/kg Q2W + Cetuximab	77	1: 100 (77)	3.6 mo. (3.3 to 3.9)	6.1% (4.2 to 9.8)	*KRAS* wt: 100 (77)
NCT02870920 [32]	2	DURVA 1500 mg Q4W + TREM 75 mgQ4W	119	-	1.8 mo. (1.8 to 1.9)	0.8% (0.2 to 1.6)	MSI-H/dMMR: 1 (1) MSS/pMMR: 98 (117) *BRAF* wt: 92 (110) *KRAS* wt: 21 (25) *BRAF* mt: 7 (8) *KRAS* mt: 78 (93)

**Table 2 cancers-13-04345-t002:** List of currently recruiting clinical trials determining clinical outcomes of PD-1/PD-L1/CTLA-4 inhibitors in patients with metastatic colorectal cancer. ICI—immune-checkpoint inhibitor; PEMBRO—pembrolizumab; NIVO—nivolumab; ATEZO—atezolizumab; IPI—ipilimumab; DURVA—durvalumab; ORR—objective response rate; PFS—progression-free survival; OS—overall survival; irAEs—immune-related adverse events; DCR—disease control rate; G—grade.

NCT	Phase	Sponsor	ICI	Primary Endpoint	Estimated Date for Primary Results
NCT04513951	2	Gruppo Oncologico del Nord-Ovest	Avelumab	PFS	May 2021
NCT04659382	2	Federation Francophone de Cancerologie Digestive	ATEZO	PFS	October 2021
NCT03983954	1	NeoTX Therapeutics Ltd.	DURVA	Incidence of irAEs ORR	November 2021
NCT03475004	2	University of Colorado, Denver	PEMBRO	ORR	December 2021
NCT03388190	2	University Hospital, Akershus	NIVO	PFS	December 2021
NCT04575922	2	Massachusetts General Hospital	NIVO + IPI	ORR	December 2021
NCT03519412	2	IFOM, the FIRC Institute of Molecular Oncology	PEMBRO	ORR	February 2022
NCT03866239	1	Hoffmann-La Roche	ATEZO	Incidence of irAEs ORR	February 2022
NCT02997228	3	National Cancer Institute (NCI)	ATEZO	PFS	April 2022
NCT03657641	2	University of Southern California	PEMBRO	Dose-limiting toxicity PFS OS	June 2022
NCT04924179	2	Huazhong University of Science and Technology	Tislelizumab	PFS	December 2022
NCT03377361	2	Bristol-Myers Squibb	NIVO	Dose-limiting toxicity Incidence of irAEs Incidence of serious irAEs Incidence of deaths ORR	January 2023
NCT04777162	2	Peking University	Tislelizumab	ORR	January 2023
NCT04730544	2	GERCOR—Multidisciplinary Oncology Cooperative Group	NIVO + IPI	Number of G.3-4 irAEs PFS	March 2023
NCT03555149	2	Hoffmann-La Roche	ATEZO	ORR	April 2023
NCT04262687	2	Federation Francophone de Cancerologie Digestive	PEMBRO	ORR	September 2023
NCT03374254	1	Merck Sharp & Dohme Corp.	PEMBRO	Dose-limiting toxicity ORR	November 2023
NCT04963283	2	University of Colorado, Denver	NIVO	DCR	February 2024
NCT03642067	2	Sidney Kimmel Comprehensive Cancer Center at Johns Hopkins	NIVO	ORR	February 2024
NCT04776148	3	Merck Sharp & Dohme Corp.	PEMBRO	OS	March 2024
NCT04017650	2	M.D. Anderson Cancer Center	NIVO	ORR Incidence of G.3-4 irAEs	June 2024
NCT03396926	2	University of California, San Francisco	PEMBRO	Dose-limiting toxicity ORR	January 2025
NCT04008030	3	Bristol-Myers Squibb	NIVO or NIVO + IPI	PFS	August 2025
NCT04430985	2	Dorte Nielsen	IPI	Disease-free survival	September 2025

## Data Availability

All data can be obtained from the corresponding author on a reasonable request via e-mail (Golo.Ahlenstiel@health.nsw.gov.au).

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
