# Peer review of "Immune-Checkpoint Inhibitors for Metastatic Colorectal Cancer: A Systematic Review of Clinical Outcomes"

_cancers, 2021, doi:10.3390/cancers13174345_

Round 1

Reviewer 1 Report

This is an excellent review on immune-checkpoint inhibitors for metastatic colorectal cancer and I commend the authors on their work.

This will be a very useful review for researchers in the field.

One minor point though is that the version of the AJCC guidelines is not mentioned for Figure 1 (e.g 8th) nor is it cited in the references. This should be amended.

Author Response

This is an excellent review on immune-checkpoint inhibitors for metastatic colorectal cancer and I commend the authors on their work. This will be a very useful review for researchers in the field.

  • One minor point though is that the version of the AJCC guidelines is not mentioned for Figure 1 (e.g 8th) nor is it cited in the references. This should be amended.

Thank you for the valuable comment. This has been amended and cited in the references

Reviewer 2 Report

Reviewer comments and suggestions

In the present study, the authors presented a systematic review that assessed the response and survival rates in patients with mCRC treated with immune-checkpoint inhibitors (ICIs). Few points need to be added to the manuscript so that it could be easier for the common reader to understand your manuscript.

Below are the comments for this paper to be incorporated in the revised version of the manuscript. 

  1. Line 19 high microsatellite instability (better to explain it here)
  2. Line 36-37 what about combinational therapy that is also needed to mention in the conclusion
  3. Figure 2 legend anti-PD-L1 need to describe here
  4. The author need to mention in the introduction about cancer patients with high microsatellite instability
  5. Line 179-181 Please describe it more based on this section
  6. Line 193-195 the author need to point his inference
  7. Line 209-210 the above matter need to be extensibility describe here
  8. It is important to establish the differences between stable and instable MS
  9. Line 224-226 what does it mean
  10. Line 246-248 is there was any prominent reason for this
  11. Line 258-260 no need of including the sentence in limitations
  12. Line 279 is there was any specific reason for not showing the clinical effect (reference) need to put
  13. Check the references 20 and 21
